# The Impact of the Course of Disease before Botulinum Toxin Therapy on the Course of Treatment and Long-Term Outcome in Cervical Dystonia

**DOI:** 10.3390/toxins13070493

**Published:** 2021-07-16

**Authors:** Harald Hefter, Isabelle Schomaecker, Max Schomaecker, Dietmar Rosenthal, Sara Samadzadeh

**Affiliations:** Department of Neurology, Moorenstrasse 5, D-40225 Düsseldorf, Germany; I.Schomaecker@gmail.com (I.S.); MS2368@cornell.edu (M.S.); dietmar.rosenthal@med.uni-duesseldorf.de (D.R.); sara.samadzadeh@yahoo.com (S.S.)

**Keywords:** cervical dystonia, natural history, course of disease, CoDB-graph, botulinum toxin therapy, long-term outcome, patient’s drawing

## Abstract

This study analyses the influence of the course of the disease of idiopathic cervical dystonia (CD) before botulinum toxin (BoNT) therapy on long-term outcomes. 74 CD-patients who were treated on a regular basis in the botulinum toxin outpatient department of the University of Düsseldorf and had received at least 3 injections were consecutively recruited after written informed consent. Patients were asked to rate the amount of change of CD in relation to the severity of CD at begin of BoNT therapy (IMPQ). Then they had to draw the course of disease of CD from onset of symptoms until initiation of BoNT therapy (CoDB-graph) on a sheet of paper into a square of 10 × 10 cm^2^ size. Remaining severity of CD was estimated by the treating physician using the TSUI-score. Demographical and treatment related data were extracted from the charts of the patients. Depending on the curvature four different types of CoDB-graphs could be distinguished. Time to BoNT therapy, increase of dose and improvement during BoNT treatment were significantly (*p* < 0.05) different when patients were split up according to CoDB-graph types. The lower the age at onset of symptoms, the shorter was the time to therapy (*p* < 0.02). Initial dose (*p* < 0.04) and actual dose (*p* < 0.009) were negatively correlated with the age of the patients at recruitment. The course of disease of CD before BoNT therapy has influence on long-term outcome. This has implications on patient management and information on the efficacy of BoNT treatment.

## 1. Introduction

Cervical dystonia (CD) is the most frequent form of focal dystonia [1,2,3,4], has a prevalence around 3 to 28/100,000 people in Europe and the U.S. [3,5,6,7], and goes along with abnormal head and neck posture and abnormal voluntary and involuntary head movements [8,9]. Since CD is a long-term disease, it requires long-term treatment [7]. Repetitive intramuscular injections of botulinum toxin (BoNT) every 10 to 16 weeks have become treatment of choice for this chronic neurological disorder [3,7,10,11].

There is overwhelming evidence that BoNT injections are an effective and save treatment of CD [3,12,13,14,15]. Nevertheless, between 17.9 and 46% of BoNT-treated CD-patients discontinue this effective treatment [13,16,17,18,19,20]. The reasons for discontinuation are poorly understood [21]. As in children with cerebral palsy (CP) [22] primary non-response [17,23], disease progression [24,25,26,27], cost of therapy, inconvenience, side effects and incorrect diagnoses [21,22,28] contribute to discontinuation of BoNT therapy in CD. Special types of CD as antecollis are associated with moderate to severe side-effects and high withdrawl rates [24,29,30]. In children with CP secondary non-response was associated with neutralizing antibody (NAB) formation in up to 75% [22]. This is consistent with recent cross-sectional studies in adult CD-patients showing a non-linear increase of NAB-formation with duration of therapy [31,32,33]. Nevertheless, the current opinion seems to be that “NAB formation occurs only in a small percentage” of BoNT/A treated patients and does not explain secondary non-response and discontinuation of BoNT therapy in the majority of CD-patients. [21,27,34,35].

Therefore, it is clinically relevant to look for other factors leading to dissatisfaction of BoNT-treated patients and discontinuation of BoNT-therapy [21,36]. It is well-known that patients starting with a simple rotational type of CD may develop further symptoms and a segmental dystonia [26,37,38,39]. The pattern of complexity of CD may increase with duration of disease even under continuous BoNT treatment [19,26]. When patients were split-up into a subgroup of patients with a good effect of BoNT therapy and a second subgroup of patients with an unsatisfactory effect, patients with an increase of the pattern of complexity of CD were more frequently found in this second subgroup of patients with an “unsatisfactory effect” (for details see [19]). Development from focal to multifocal or segmental dystonia in up to 30% of CD-patients has been reported [26,37,38,39], but generalization does not occur in the majority of adult patients and is a red flag for special, genetically determined or symptomatic dystonia [39,40,41].

In a recent study on the course of severity of CD under BoNT/A therapy it was demonstrated that new symptoms (as pain, myoclonic jerks, tremor or loss of voluntary head control) may occur and expand the spectrum of symptoms of CD during BoNT-therapy despite of improvement of head position [26]. Furthermore, it was demonstrated that the duration of the time from onset of symptoms to onset of therapy had a negative impact on the outcome of long-term BoNT therapy [26]. That was comparable to pallidal deep brain stimulation of primary dystonia where the duration of disease before operation was the only parameter showing a significant (negative) correlation with outcome (for details see [42,43]).

Furthermore, it has been reported that the majority of patients with a low initial severity of CD at initiation of botulinum toxin therapy did not experience an improvement, at least not during the first 3 to 4 treatment cycles [44]. 

But we were not only interested to analyze the influence of the time to therapy or of the initial severity of CD at begin of BoNT therapy on later outcome, but also to study the influence of the entire course of disease on the course of treatment and the long-term treatment related data. A closer look to patients’ charts and the literature was rather disappointing. In most of our patients’ charts the pre-BoNT history was not mentioned or not described in detail. In the literature only little information was available on the course of CD before BoNT therapy. Three different phases of the spontaneous course of CD have been described by neurosurgeons [45,46]. Deterioration is observed during the first 5 years, stable condition during the next 5 years and possible improvement thereafter [33]. In a recent review remission in dystonia was analyzed [46]. Spontaneous complete remissions seem to be rare (3 to 15%) and usually occur during the first 3 to 5 years after first onset symptoms [27,45,47]. Patients with remissions were significantly younger (about 7.1 years in the mean) at the onset of symptoms than patients without remissions [47]. After initiation of BoNT therapy symptoms may fluctuate considerably. 

To study the influence of patient’s pre-BoNT history on long-term outcome after BoNT in more detail, a new approach to this problem was used. Patients had to draw the course of disease of CD prior to BoNT therapy (CoDB-graph). Digitization of these graphs allowed to compare and correlate the CoDB-graphs with treatment related data and the long-term outcome of BoNT-therapy. To our knowledge this is the first time that patients’ perception and drawing of the course of disease is used for comparison with physician’s scoring of symptoms and treatment related data. 

## 2. Results

### 2.1. Demographical Data, Treatment Related Data and Outcome of the Entire Cohort

Demographical, treatment related data and outcome of the entire cohort are presented in the column “All” of Table 1. The female/male ratio was 1.9 (for details see Table 1). Patients had a typical mean AOS of 45.3 years, a mean DURS of 68.9 months (=5.74 years) with an extremely broad range (1 to 438 months), and a mean ITSUI of 8.9.

Under BoNT/A treatment, the TSUI-score highly significantly (*p* < 0.001) improved from 8.9 by about 50% (mean IMPTSUI = 4.5) to 4.4. (ATSUI). The improvement assessed by the patients was 42.9% or 46% in dependance on the use of a questionnaire (IMPQ) or a visual analogue scale (IMPD) (for details see Table 1).

Mean IDOSE was rather low (166 uDU). Dose was increased by 51 uDU (INDOSE) up to 218 uDU (ADOSE) during a mean DURT of 115.7 months (=9.64 years). 

### 2.2. Correlation Analysis of Demographical and Treatment Related Data

The earlier the onset of symptoms the shorter was the time to treatment (AOS/DURS: r = 0.281; *p* < 0.02). Younger patients were treated with higher doses (AGE/IDOSE: r = −0.253; *p* < 0.04; AGE/ADOSE: r = −414; *p* < 0.009). There was a highly significant correlation between actual TSUI and actual total dose (ATSUI/ADOSE: r = 0.497; *p* < 0.009), but not between ITSUI and IDOSE (ITSUI/IDOSE: r = 0.12; n.s.). IDOSE was highly significantly correlated with ADOSE (IDOSE/ADOSE: r = 0.677; *p* < 0.009), but not ITSUI with ATSUI (ITSUI/ATSUI: r = 0.011; n.s.). Increase of dose was highly significantly correlated with ADOSE (INDOSE/ADOSE: r = 0.652; *p* < 0.009) and with ATSUI (INDOSE/ATSUI: r = 0.326; *p* < 0.01), but not with ITSUI (INDOSE/ITSUI: r = −0.050; n.s.).

The two estimations of improvement of the patients (IMPQ and IMPD; for details see Methods) were highly correlated (IMPQ/IMPD: r = 0.951; *p* < 0.009). Estimation of improvement by the treating physician using the TSUI-score was correlated with IMPQ (IMPQ/IMPTSUI: r = 0.749; *p* < 0.009). IMPTSUI was highly significantly negatively correlated with ATSUI (IMPTSUI/ATSUI: r = −0.709; *p* < 0.009) and positively with ITSUI (IMPTSUI/ITSUI: r = 0.601; *p* < 0.009). Also, IMPQ and IMPD were significantly negatively correlated with ATSUI (IMPQ/ATSUI: r = −0.437; *p* < 0.009; IMPD/ATSUI: r = −0.340; *p* < 0.01), but not with ITSUI (for details see Table 2).

### 2.3. Three Main and One Exceptional CoDB-Graph Type 

The main purpose of the present study is to analyze the course of disease (CoD) before BoNT/A therapy and whether patients with different CoDs are treated differently. To achieve this purpose patients had to draw CoD on a sheet of paper into a square of 10 × 10 cm^2^ size. Three examples of original CoDB-graphs (before digitization) are presented in Figure 1.

Most of the patients (71 out of 74) succeeded to perform a CoDB-graph which could be digitized with sufficiently high quality for further analysis (for details see Materials and Methods). Sixty-nine of the 71 digitized CoDB-graphs could be classified according to the main curvature: 16 (23%) patients had drawn a graph with a rapid onset followed by an increasingly slower progression (RO-type) (Figure 1 left side), 23 (32%) patients a graph with a continuous disease progression (CO-type) (Figure 1 middle part) and 30 (42%) patients a graph with an initially slow progression followed by a delayed faster and faster progression (DO-type) (Figure 1 right side). A typical example for each of these three main CoDB-graph types (after digitization) is presented in Figure 2 (left side). All graphs of a special type are presented on the right side of Figure 2. In the middle column of Figure 2 the average CoDB-graph of a special type together with the corresponding standard deviation range is presented. 

Only two patients (<3%) had drawn a CoDB-graph which could not be classified as RO-, CO- or DO-type. One patient had experienced a remission over several years, the other patient no disease progression over a long time (Figure 3). These two patients and their CoDB-graphs were excluded from further analysis.

### 2.4. Comparison of Patient-Subgroups Classified According to the CoDB-Graph Type

Patients were subdivided into a RO-, CO- and DO-subgroup depending on the type of CoDB-graph they had produced. The distribution of males and females was different in the RO-, CO- and DO-subgroup, but the chi-square testing failed to be significant (*p* = 0.13; n.s.). We also performed a two group ANOVA (with all the parameters used in Table 1) to compare females and males, but for none of the 12 parameters a significant difference could be found after Bonferroni-Holm correction.

The three group ANOVA yielded significant “between group differences” for DURS (*p* < 0.01), INDOSE (*p* < 0.05) and IMPD (*p* < 0.02), but not for the initial severity (ITSUI) and the final outcome (ASTUI) and final total dose (ADOSE). Trends (*p* < 0.15) were observed for IDOSE and patient’s assessment of improvement using a questionnaire (IMPQ; for details see Table 1). 

Because of the difference in the course of treatment, we describe the three groups in more detail. In the RO-group AOS and DURS was the lowest, DURT the longest, ITSUI the highest, IDOSE the highest, INDOSE the lowest and IMPD the highest (see Table 1). 

In the DO-group DURS was the longest, ITSUI as high as in the RO-group, INDOSE and ADOSE and IMPTSUI the highest with an extremely broad variation (see Table 1). 

In the CO-group the percentage of females and the age at onset was the highest, ITSUI, IDOSE, IMPQ, IMPD and IMPTSUI the lowest.

In summary, the three different main types of the course of disease before BoNT therapy had influence not only on the initiation of BoNT/A therapy, but also on dose adjustment and long-term improvement. 

## 3. Discussion 

### 3.1. General Remark on CoDB-Graph Drawing

The CoDB-graphs had to be produced by drawing a continuous graph from the left lower corner of a square to the right upper corner. One might argue that there are not many principally different possibilities to perform such a graph: a linear, a convex, a concave and a waxing and waning one. Indeed, these were the 4 different types observed in the present study. However, if patients had chosen deliberately a special type of these 4 graph-types we would have expected that the linear CO-type had been drawn most frequently since this is the easiest type to draw. But the most frequent type was the DO-type.

Having demonstrated that patients draw different types of CoDB-graphs, the relevant question is, which we will discuss in the following, whether clinical information can be derived when patients were grouped according to their graph-type. 

### 3.2. The Frequency of the Four Different CoDB-Graph Types

Four different types of CoDB-graphs could be distinguished. The most frequent type (42%) was the DO-type with a mild onset, but an increasing severity and a final rapid deterioration just before begin of BoNT therapy. Time to therapy (DURS) was longer than 8.6 years in the mean. The second frequent type (32%) was the CO-type with a mean DURS of less than 4 years. In about 1/4th of the patients a severe rapid manifestation of CD was experienced followed by little further progression (RO-type) with a short DURS of less than 3 years in the mean (for details see Table 1). 

Only 2 (<3%) CoDB-graphs did not fit into the classification by these three types. These two patients had experienced a clear relapse or no progression over years (Figure 3). This is consistent with the literature that relapses of CD were infrequent (<15%) and occur during the first five years after manifestation [21,45,47]. 

As mentioned above the frequencies of the 4 different CoDB-graph types is a first hint that the type drawn by a patient was not chosen deliberately. Observation of patients during drawing revealed that patients were eager to produce a “correct” image of their CoD. The different frequencies of the 4 CoDB-graph types demonstrate that patients did not try to finish this drawing task the easiest way by drawing a simple straight line. Only 1/3rd of the patients decided to draw the CO-type. The majority of the patients had drawn a more complex curve. We therefore were convinced that the CoDB-graphs contained useful clinical information.

None of the patients had drawn a course of disease which had previously been described with a deterioration during the first 5 years, followed by a stable condition during the next 5 years and a possible improvement thereafter [46]. Such a CoD may have been typical for patients in the pre-BoNTera in patients who did not undergo brain surgery. 

### 3.3. Clinical Differences between the Three RO, CO- and DO-Patient Subgroups

In the entire cohort the female/male ratio was 1.9 as in other larger samples of patients with idiopathic CD [1,2,3]. There was a clear trend (*p* = 0.13) that the distribution of females and males was different in the three different patient subgroups. The percentage of males was considerably higher in the RO-group (43.8%) and in the CO-group (40%) than in the DO-group (17.4%). 

Interestingly, ITSUI was about the same in the RO-, CO- and DO-subgroup. Obviously, there seems to be a certain limit of severity, patients can tolerate. Beyond this limit of severity patients look for help and are referred to botulinum toxin therapy. In the RO-group this limit was reached within a short time (<3 years), in the DO-group it took more than 8 years until patients were referred for BoNT treatment. The time to therapy (DURS) was significantly (*p* < 0.01) different between the three patient subgroups.

### 3.4. Differences in the Initiation of BoNT Treatment across the RO-, CO-, and the DO-Group

The dose chosen at the beginning of BoNT therapy was highly significantly (*p* < 0.009) correlated with the actual dose which was applied years later. This indicates that the choice of the dose at the beginning of BoNT therapy has implications for the entire course of treatment.

IDOSE did not reveal a significant correlation with ITSUI. Obviously, the treating physicians did not decide to inject the patients only in dependance on the severity of abnormal head position. There was a clear tendency (*p* = 0.14) that IDOSE was different in the RO-, CO-, and DO-group. RO-patients with a rapid severe manifestation of CD and the shortest time to therapy were treated with the highest initial doses (205 uDUs in the mean), whereas patients in the DO-group with a much longer time to therapy but with the same initial severity were treated with much lower doses (161 uDUs in the mean). In patients of the CO-group who had drawn a continuous worsening, BoNT-treatment was initiated with the lowest doses.

Obviously, treating physicians did not only take into account the severity of CD when the patients presented the first time to be treated with BoNT, but also reacted to the course of disease. This is probably the reason, why patients with a short DURS but a low ITSUI in the CO-group were treated with the same low IDOSE as the DO-group with a much longer DURS but a higher ITSUI. This is also consistent with the use of high IDOSEs in the RO-group. Whether treating physicians had been aware of this or had reacted by intuition remains unclear. Nothing was documented on the course of disease in the charts. In some of the charts age at onset of symptoms was documented, but nothing concerning the course of CD.

### 3.5. Differences in Dose Adjustment during BoNT Treatment across the RO-, CO-, and the DO-Group

Although there had been a trend to differences in initial dose, no difference (*p* = 0.65) could be detected between the doses used at the time of recruitment (ADOSE; see Table 1) after more than 9 years of BoNT-treatment. Across all three patient subgroups there was a highly significant (*p* < 0.009) correlation between the actual severity of CD (ATSUI) and the actual dose. During the course of BoNT treatment the treating physicians adjusted the dose in dependance on the severity of CD. They tried to improve CD by using higher doses. This is obvious from the significant correlations between increase of dose and ATSUI (*p* < 0.01) and between INDOSE and ADOSE (*p* < 0.009).

In the RO-group with higher initial doses only a small increase of dose (7 uDUs) was made, although the duration of treatment was longer than 12 years. In the two subgroups which started with a low initial dose the dose was considerably increased during treatment: 65 uDU in the CO- and 73 uDU in the DO-group. Increase of dose was the second parameter which was significantly (*p* < 0.05) different between the RO-, CO-, and DO-group.

### 3.6. Differences in Outcome between the RO-, CO- and DO-Patient Subgroups

The actual TSUI-score (ATSUI) did not significantly differ between the three groups. Patients in the DO-group who were treated with the highest doses (239 uDUs in the mean) had the lowest remaining severity of CD (ATSUI = 4.2). The improvement rated by the treating physicians (IMPTSUI) varied between 47% (CO-group) to 55% (DO-group) and did not show any difference between the groups. In the end, dose adjustments by the treating physicians had been successful. After more than 9 years of BoNT therapy mean TSUI-score of all three patient groups was within the 4 to 5 TSUI-score range. This has repeatedly been reported to be the mean outcome in long-term treated CD-patients in our institution [48,49]. 

However, when patients rated the improvement of CD by means of a questionnaire (IMPQ) a tendency (*p* = 0.10) to differences between the subgroups was found, IMPQ ranged from less than 36% in the CO-group to more than 56% in the RO-group (see Table 1). 

Rating of the patients of the remaining severity of CD (ASCD) in percent of the severity of CD at onset of BoNT therapy (using the right edge of the CoDA-graph as a visual analogue scale) yielded a remarkably low value of about 34% in the RO-group corresponding to an improvement (IMPD) of 66% in the mean. This was better than the improvement scored by the physicians (IMPTSUI = 50%). In the other two groups IMPD was smaller than IMPTSUI (37% vs. 47% in the CO-group; 41% vs. 55% in the DO-group. This is the typical situation, that physicians rate the improvement of their therapy better than the patients (for details see [35,36]). 

IMPD was the third parameter which showed a significant (*p* < 0.02) difference between the three patient subgroups.

## 4. Concluding Remarks

Out of 74 participants in the present study 71 (=95.9%) CD-patients succeeded to produce a CoDB-graph which could be used for further analysis. This demonstrates that CoD-graph drawing can be used in clinical practice. Three different main CoDB-graphs could be distinguished. When patients were split-up according to these three main graph-types significant differences between these 3 patient subgroups in treatment related data and outcome were detected. 

Treating physicians had reacted to the differences in history of the patients prior to BoNT treatment and treated the patients not only in dependance of severity of CD. This then had implications on the entire course of BoNT treatment. Interestingly, the outcome rated by the physician did not differ between the three patient groups. But outcome assessed by the patients was significantly different. Further studies are necessary to analyse whether a treatment strategy optimizing patient’s satisfaction or a treatment strategy optimizing head position is the better long-term patient management in CD.

Patients did not only draw CoDB- but also CoDA-graphs as a possibility to monitor BoNT therapy. Questions are whether also different types of responses to BoNT/A (CoDA-graphs) can be distinguished, whether the CoDA-graphs can be classified in a similar way as the CoDB-graphs and whether a special type of CoDB-graph is associated with a special type of CoDA-graph. These questions will be addressed in a subsequent paper. 

It is quite obvious, that CoD-graph drawing is a universal method and not limited to the analysis of patients with CD or treatment with botulinum toxin. It may become even more relevant in disease entities as e.g., multiple sclerosis where therapy is highly dependent on the course of disease.

## 5. Strengths and Limitations of the Study

To our knowledge this is the first study which digitizes and analyses the course of disease drawn by the patients. It underlines the importance of patient’s history before BoNT therapy on the long-term outcome. But it has to be taken into account that in the present study only those patients were recruited to draw a CoDB-graph who had decided long ago to be treated with BoNT. We therefore think that the CoDs of the patients in the present cohort underly a selection bias and do not represent the CoDs of all CD-patients. To learn more about the spontaneous natural course of disease of CD also those patients should be analysed who neither undergo DBS nor BoNT treatment.

## 6. Materials and Methods 

The present study was performed according to the guidelines for good clinical practise (GCP) and according to the declaration of Helsinki. It was approved by the local ethics committee of the University of Düsseldorf (number: 4085; date: 5 April 2013; updated: 8 May 2018).

### 6.1. Patients and Treatment-Related Data 

Inclusion criteria of the study were: (i) age over 17, (ii) diagnosis of idiopathic CD, (iii) onset of therapy in the out-patient department of the University of Düsseldorf (Germany) and continuous treatment every 12 to 13 weeks without interruption of BoNT therapy of more than one treatment cycle (iv) at least 3 injections of BoNT, (v) written informed consent. Exclusion criteria were: (i) patient under legal care, (ii) multifocal, segmental and/or symptomatic dystonia at onset of BoNT therapy, (iii) additional other disabling disease than CD. Especially patients with a history of or clinical manifestation of disturbances of mood and/or perception were excluded. 

Charts of regularly treated CD-patients were screened. Patients who met the exclusion and inclusion criteria (i) to (iv) were informed on the purpose of the study. 74 patients gave informed written consent and were consecutively recruited. 

The following demographical data were extracted from the charts: age at day of recruitment (AGE), age at onset of symptoms (AOS) and age at onset of therapy (AOT). Duration of therapy (DURT= AGE − AOT) and time span during which patients had tolerated symptoms without BoNT therapy (DURS= AOT − AOS) were determined. 

Patients had to rate the change of CD since onset of BoNT-therapy in percent of the severity of CD at onset of therapy (IMPQ). The treating physician scored the actual severity of CD at the day of recruitment by means of the TSUI-score (ATSUI; [12]), documented the BoNT-preparation used as well as the actual total dose (ADOSE). TSUI-score at onset of therapy (ITSUI), initial BoNT-preparation and initial total dose (IDOSE) were extracted from the charts. For sake of comparison doses of different BoNT-preparations were transformed into unified dose units (uDU) by leaving ona- and incobotulinum-toxin (onaBoNT/A; incoBoNT/A) doses unchanged and dividing abobotulinumtoxin (aboBoNT/A) doses by 3 and rimabotulinumtoxin (rimaBoNT/B) doses by 30 following evidence-based data and a European consensus paper [3]. Improvement of severity of CD on the basis of the TSUI-score (IMPTSUI) was calculated as (ITSUI − ATSUI) × 100/ITSUI). The increase of dose (INDOSE) during treatment was calculated as ADOSE-IDOSE. 

Alternative therapies (acupuncture, physiotherapy, etc. [50]) were not controlled in the present study.

### 6.2. Drawing of the Course of Disease Graphs (CoD-Graphs)

To draw the course of severity of CD from onset of symptoms until onset of BoNT/A therapy (CoDB-graph) patients were comfortably seated in front of a desk with both hands on the desk, one hand holding a piece of paper with a square of 10 × 10 cm^2^ size and the other holding a pen. The investigator explained to the patient that the lower edge of the square (*x*-axis) should correspond to the time from onset of symptoms to onset of BoNT-therapy and the upward direction (*y*-axis) to severity of CD. The right upper corner should represent the severity of CD at onset of BoNT-therapy. The patient had to draw the CoDB-graph from the left lower corner (=0%) to the right upper corner (=100%) without interruptions. If the patient did not succeed to draw a continuous graph, but had produced several line fragments, a second continuous graph had to be drawn into a new square. Even a third attempt was allowed, but not a fourth one. Also, verbal support by the investigator was allowed, but not drawing assistance. To avoid a bias no example from another patient or the investigator was shown to the patient. An original example of a CoDB-graph is presented in Figure 1 and Figure 4.

To draw the course of severity of CD from onset of BoNT/A until the day of recruitment into the study (CoDA-graph) a second piece of paper with a square of 10 × 10 cm^2^ was presented. The right edge of the square was used as a visual analogue scale: the actual severity of CD (ASCD) had to be marked on this edge in relation to severity of CD at onset of therapy (=100%). Improvement of CD estimated by drawing of the CoDA-graph (IMPD) was calculated as (10 − ASCD) × 10. Then the CoDA-graph had to be drawn continuously from the left upper corner (=100%) to the mark of ASCD on the right edge of the square. Further analysis of the CoDA-graphs will be presented in a separate paper. 

After rating the change of severity of CD and drawing of the CoD-graphs, patients underwent a detailed clinical investigation and were injected. 

### 6.3. Classification of the CoDB-Graphs

Three out of 74 patients were unable to draw a continuous graph, even after three attempts. Their demographical and treatment related data were included in the study, but their graphs had to be excluded from the following analysis. 71 CoDB-graphs were scanned by means of a standard scanner. The CoDB-graphs were digitized using a commercially available software DIGITIZEIT^®^ (Braunschweig, Germany) after origin and end of the *x*-axis, of the *y*-axis and of the CoDB-graph had been marked. The software produced an *x*,*y*-table when a stick was moved along the scan of the CoDB-graph from the origin to the end of the graph (Figure 1 (middle part)). The digitized version of the example in Figure 1 (left side) is presented in Figure 1 (right side). 

Depending on the mean curvature of the digitized CoDB-graphs four different types of graphs were distinguished. The rapid onset (RO)-type had a negative mean curvature and revealed a rapid onset followed by the slower progression (Figure 2 (upper part)). The continuous (CO)-type did not have any curvature in the mean and showed a continuous progression from onset of symptoms until initiation of BoNT/A therapy (Figure 2 (middle part)). The delayed onset (DO)-type had a positive curvature, started with a slow progression which became continuously faster (Figure 2 (lower part)). As other (OO)-type all graphs were classified which could not be classified as RO-, CO- or DO-type (Figure 3). 

### 6.4. Statistics

Patients were split-up into 4 sub-groups (RO; CO; DO; OO). A chi-square analysis was performed whether sex distribution was different across patient subgroups. A two-group ANOVA was performed to test whether AGE, AOS, DURS, DURT, IDOSE, ADOSE, INDOSE, ITSUI, ATSUI, IMPTSUI, IMPQ and IMPD were different between females and males. A three-group ANOVA was performed to test whether the same parameters were significantly different between the RO-, CO- or DO-subgroup. The OO-subgroup contained only 2 patients and was excluded from the ANOVA-analysis. A Bonferroni-Holm correction was applied to compensate for multiple comparisons. Non-parametric rank-correlations (Spearman’s rho) were determined whether demographical and treatment related data were significantly correlated. All statistical analyses were performed using the SPSS^®^ statistics package (version 25; IBM^®^, Armonk, NY, USA).

## Figures and Tables

**Figure 1 toxins-13-00493-f001:**
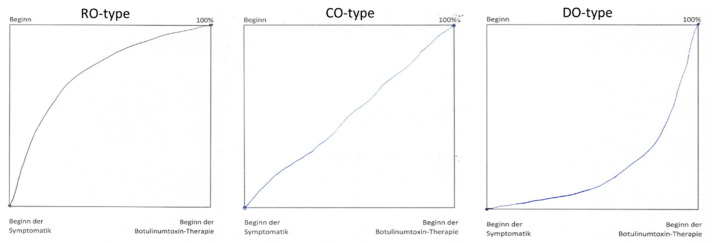
Three examples of original CoDB-graphs are presented: a rapid onset (RO)-type graph on the (**left side**), a continuous onset (CO)-type graph in the (**middle part**), and a delayed onset (DO)-type graph on the (**right side**).

**Figure 2 toxins-13-00493-f002:**
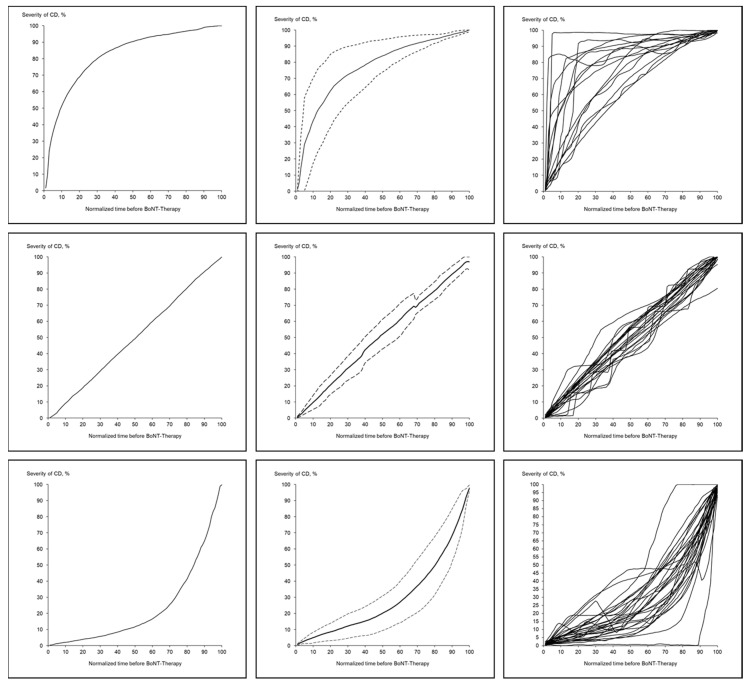
(**left upper part**): A typical CoDB-graph of the RO-type is shown; (**upper middle part**): The average graph with the standard deviation (S.D.)-range for all RO-type CoDB-graphs is presented; (**right upper part**): All 16 CoDB-graphs of the RO-type are presented; (**left middle part**): A typical CoDB-graph of the CO-type is shown; (**middle middle part**): The average graph with S.D.-range of all CO-type CoDB-graphs is presented; (**right middle part**): All 30 CoDB-graphs of the CO-type are presented; (**left lower part**): A typical CoDB-graph of the DO-type is shown; (**middle lower part**): The average graph with the S.D.-range for all DO-type CoDB-graphs is presented; (**right lower part**): All 23 CoDB-graphs of the DO-type are presented.

**Figure 3 toxins-13-00493-f003:**
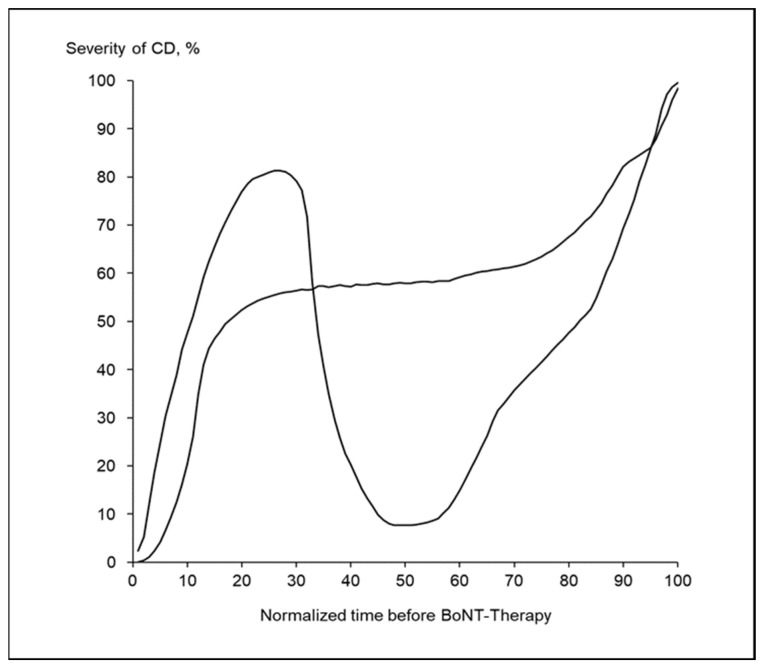
Two exceptional CoDB-graphs of the OO-type are presented. One plot reveals an example of a patient who reports a clear remission over years. The other plot was drawn by a patient who reported a long-time span without progression of CD for years.

**Figure 4 toxins-13-00493-f004:**
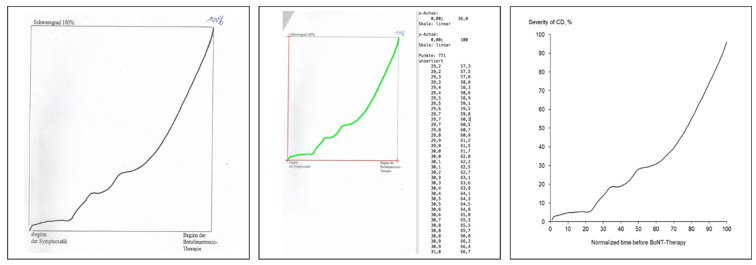
(**left side**): The scan of a typical CoDB-graph of a 52-year-old female patient is shown. (**middle part**): Digitization of the scan in Figure 1 (left side) yields an *x*,*y*-table which can be used for further analysis. (**right side**): The *x*,*y*-plot of the digitized data demonstrates the similarity to the original drawing.

**Table 1 toxins-13-00493-t001:** Demographical and treatment related data for all three subgroups and the entire cohort.

Parameter		RO	CO	DO	ALL	Significance-Level
n		16	23	30	74	
female/male		9/7	19/4	18/12	49/25	0.13; n.s.
AGE (years)	MV/SD	57.0/8.6	61.0/13.3	61.1/12.1	60.2/11.6	0.49; n.s.
	MIN–MAX	41.7–76.5	28.7–81.3	43.4–87.0	28.7–87.0	
AOS (years)	MV/SD	42.9/11.6	48.8/12.5	43.6/12.9	45.26/12.5	0.24; n.s.
	MIN–MAX	21.4–61.8	26.8–73.4	14.7–64.6	14.7–73.4	
DURS (months)	MV/SD	34.0/29.3	46.1/74.0	105.2/114.5	68.9/91.5	*p* < 0.01
	MIN–MAX	3.0–97.0	1.0–324.2	2.0–438.2	1.0-438.2	
DURT (months)	MV/SD	144.2/101.6	102.3/72.6	107.8/71.1	115.7/80.3	0.25; n.s.
	MIN–MAX	11.05–321.2	6.1–270.3	0.6–282.3	0.6–321.2	
IDOSE (uDU)	MV/SD	205.2/96.5	155.2/62.8	160.6/80.1	166.3/80.6	0.14; n.s.
	MIN–MAX	35.0–450.0	31.3–300.0	75.0–500.0	31.3–500.0	
ADOSE (uDU)	MV/SD	212.3/112.7	212.8/108.5	238.8/116.2	217.9/114.7	0.65; n.s.
	MIN–MAX	50.0–450.0	40.0–400.0	95.0–500.0	15.0–500.0	
INDOSE (uDU)	MV/SD	7.2/49.9	64.8/77.4	73.2/100.1	50.6/87.2	*p* < 0.05
	MIN–MAX	−100.0–100.0	0.0–275.0	−50.0–275.0	−110.0–275.0	
ITSUI	MV/SD	9.2/1.8	8.3/1.9	9.2/3.1	8.9/2.4	0.75; n.s.
	MIN–MAX	8.0–12.0	5.0–10.0	4.0–13.0	4.0–13.0	
ATSUI	MV/SD	4.7/3.1	4.5/2.5	4.2/2.6	4.4/2.6	0.83; n.s.
	MIN–MAX	0.0–10.0	0.0–10.0	0.0–9.0	0.0–10.0	
IMPTSUI	MV/SD	4.6/3.1	3.7/2.7	5.1/5.0	3.9/3.8	0.37; n.s.
	MIN–MAX	0.0–10.0	0.0–8.0	−2.0–13.0	−2.0–13.0	
IMPQ	MV/SD	56.7/30.5	35.7/26.4	40.8/33.6	42.9/31.3	0.10; n.s.
	MIN–MAX	0.0–90.0	−20.0–90.0	−30.0–90.0	−30.0–90.0	
IMPD	MV/SD	66.0/29.9	37.0/25.0	41.0/35.0	46.0/32.0	*p* < 0.02
	MIN–MAX	8.0–98.0	−20.0–75.0	−30.0–92.0	−30.0–98.0	

RO = rapid onset subgroup; CO = continuous onset subgroup; DO = delayed onset subgroup (for details see Methods); MV = mean value; SD = standard deviation; uDU = unified dose units; AGE = age at recruitment; AOS = age at onset of symptoms; DURS = time from onset of symptoms to BoNT therapy; DURT = duration of BoNT therapy; IDOSE = dose at onset of BoNT therapy; ADOSE = dose at investigation (actual dose); INDOSE = increase of dose during BoNT therapy; ITSUI = TSUI-score at onset of BoNT therapy; ATSUI = TSUI at investigation (actual TSUI); IMPTSUI = improvement according to TSUI-score; IMPQ = improvement of symptoms according to questionnaire; IMPD = improvement according to drawing (for details see Methods).

**Table 2 toxins-13-00493-t002:** Correlations of demographical, treatment related data and outcome of the entire cohort.

	AGE	AOS	DURS	DURT	IDOSE	ADOSE	INDOSE	ITSUI	ATSUI	IMPQ	IMPD	IMPTSUI
AGE	.	0.654	0.264	0.257	−0.253	−0.414	−0.230	0.170	−0.010	0.070	0.090	0.012
AOS	0.009	.	−0.281	−0.244	−0.190	−0.297	−0.130	−0.180	−0.070	0.120	0.140	−0.100
DURS	0.030	0.020	.	0.242	−0.180	−0.200	−0.210	0.150	0.220	−0.230	−0.230	−0.090
DURT	0.030	0.040	0.050	.	0.263	−0.010	−0.080	0.070	0.050	0.180	0.200	0.017
IDOSE	0.040	.	.	0.040	.	0.677	0.020	0.120	0.376	0.140	0.100	−0.100
ADOSE	0.009	0.020	.	.	0.009	.	0.652	0.080	0.497	0.030	−0.040	−0.018
INDOSE	.	.	.	.	.	0.009	.	−0.050	0.326	−0.040	−0.140	−0.100
ITSUI	.	.	.	.	.	.	.	.	0.011	0.400	0.360	0.601
ATSUI	.	.	.	.	0.009	0.009	0.010	.	.	−0.437	−0.340	−0.709
IMPQ	.	.	0.050	.	.	.	.	.	0.009	.	0.951	0.749
IMPD	.	.	.	.	.	.	.	.	0.010	0.009	.	0.020
IMPTSUI	.	.	.	.	.	.	.	0.009	0.001	0.009	.	.

Numbers above the diagonal are correlation coefficients, numbers below the diagonal indicate significance-levels below 0.05. AGE = age at recruitment; AOS = age at onset of symptoms; DURS = time from onset of symptoms to BoNT therapy; DURT = duration of BoNT therapy; IDOSE = dose at onset of BoNT therapy; ADOSE = dose at investigation (actual dose); INDOSE = increase of dose during BoNT therapy; ITSUI = TSUI-score at onset of BoNT therapy; ATSUI = TSUI at investigation (actual TSUI); IMPQ = improvement of symptoms according to questionnaire; IMPD = improvement according to drawing; IMPTSUI = improvement according to TSUI-score; (for details see Methods).

## Data Availability

Data available on request due to restrictions eg privacy or ethical. The data presented in this study are available on request from the corresponding author.

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
