# Peer review of "The Impact of the Course of Disease before Botulinum Toxin Therapy on the Course of Treatment and Long-Term Outcome in Cervical Dystonia"

_toxins, 2021, doi:10.3390/toxins13070493_

Round 1

Reviewer 1 Report

The manuscript entitled "The impact of the course of disease before botulinum toxin therapy on the course of treatment and long-term outcome in cervical dystonia" is well written and worthy of publication.

I would like to suggest to the authors the following improvements:

Abstract

Page 1. Line 6. Please move the acronym (BoNT) before the term therapy.

Introduction

Page 1. Lines 39-44. Please consider the opportunity to improve the punctuation to make the text easier for the reader.

Page 2. Line 62. Please close the square bracket before the number 22.

Results and Discussion

I would like to suggest to the authors to consider combining results and discussion to make it easier to understand the paper.

Page 8. Lines 255-260. I would like to suggest to the authors improving this paragraph explaining in depth the statement reported in the last phrase, "In summary, […]".

Page 8. Lines 261-269. The authors should consider improving this section by moving the paragraph on page 8, lines 261-269 because this represents a study's weakness.

Conclusion and Perspective. The authors should consider the opportunity to fuse these two sections as a new entitled as "Concluding remarks".

Strengths and Limitations of the study. I appreciate a lot this section and would like to suggest improving it, also detailing the use of the course of the disease graph and its importance for:

  • the present study
  • the understanding of some illness courses
  • decision making in clinical practices.

Author Response

The manuscript entitled "The impact of the course of disease before botulinum toxin therapy on the course of treatment and long-term outcome in cervical dystonia" is well written and worthy of publication.

I would like to suggest to the authors the following improvements:

Abstract

Page 1. Line 6. Please move the acronym (BoNT) before the term therapy.

Introduction

Page 1. Lines 39-44. Please consider the opportunity to improve the punctuation to make the text easier for the reader.

Page 2. Line 62. Please close the square bracket before the number 22.

Results and Discussion

I would like to suggest to the authors to consider combining results and discussion to make it easier to understand the paper.

Page 8. Lines 255-260. I would like to suggest to the authors improving this paragraph explaining in depth the statement reported in the last phrase, "In summary, […]".

Page 8. Lines 261-269. The authors should consider improving this section by moving the paragraph on page 8, lines 261-269 because this represents a study's weakness.

Conclusion and Perspective. The authors should consider the opportunity to fuse these two sections as a new entitled as "Concluding remarks".

Strengths and Limitations of the study. I appreciate a lot this section and would like to suggest improving it, also detailing the use of the course of the disease graph and its importance for:

the present study

the understanding of some illness courses

decision making in clinical practices.

Is corrected now.

We have modified this sentence.

The missing bracket is added.

The usual journal policy is to keep results and discussion separately, but we agree that it is sometimes better to have the results available in the discussion. We therefore now repeat some results in the discussion.

This paragraph is now modified.

This paragraph has also been modified and moved to the limitations section.

These two sections are fused to a new section:

“Concluding remarks”.

We are thankful for these suggestions and have emphasized these three different aspects.

Reviewer 2 Report

In this study the authors analysed the influence of the course of the disease of idiopathic cervical dystonia (CD) before botulinum toxin therapy (BoNT) on long-term outcomes. For this, 74 CD-patients who were treated on a regular basis in the botulinum toxin outpatient department of the University of Düsseldorf and had received at least 3 injections were consecutively recruited after written in-formed consent. All patients were asked to rate the amount of change of CD in relation to the severity of CD at onset of BoNT therapy (IMPQ). Then they had to draw the course of disease of CD from onset of symptoms until onset of BoNT therapy (CoDB-graph) into a square of 10 × 10 cms. As a result, the course of disease of CD before BoNT therapy has influence on long-term outcome.

The results of the study are new and interesting. Nevertheless, this manuscript needs substantial improvements and corrections before publishing may be possible.

General points:

Please do your References list at the end of the manuscript according to “Toxins”.  

Please add as a Figure an original drawing of 3 different patients.

Special points:

Keywords: please add also to keywords: CoDB-graph; patient’s drawing

Introduction

The Introduction section should be substantially improved.

You said:  Cervical dystonia (CD) is the most frequent form of focal dystonia

[1]..

Please add more references at the end of this sentence.

You said: Repetitive intramuscular injections of botulinum toxin (BoNT) every 10 to 16 weeks have become treatment of choice for this chronic neurological disorder [6].

Please add more references at the end of this sentence.

You said:

Nevertheless, up to 46% of BoNT-treated CD-patients discontinue this effective treatment [11]).

Please add more references at the end of this sentence.

You said: It is well-known that CD may progress from a simple type to segmental dystonia [14,15].

Please describe these two studies exactly.

You said: The pattern of complexity of CD may increase with duration of disease even under continuous BoNT treatment [16]. When patients were split-up into a subgroup with an unsatisfactory and into another subgroup of patients with a good effect of BoNT therapy patients with an increase of the pattern of complexity of CD were more frequently found in the subgroup of patients with an “unsatisfactory effect” (for details see [16]). Development from focal to multifocal or segmental dystonia in up to 30% of CD-patients has been reported [14,15], but generalization is a red flag for symptomatic dystonia [17].

Please describe all these studies exactly.

You said: In a recent study on the course of severity of CD it was demonstrated that new symptoms may occur and expand the spectrum of symptoms of CD during BoNT-therapy despite of improvement of head position [18]. Furthermore, it was demonstrated that the duration of the time from onset of symptoms to onset of therapy had a negative impact on long-term BoNT therapy [18]. That was comparable to pallidal deep brain stimulation of primary dystonia where the duration of disease before operation was the only parameter showing a significant (negative) correlation with outcome (for details see [19,20]).  

Furthermore, it has been reported that the initial severity of CD at onset of botulinum toxin therapy has influence on long-term outcome, at least during the first 3 to 4 treatment cycles [21]. But we were not only interested to analyse the influence of the time to therapy or the severity of CD at onset of BoNT therapy on later outcome, but also to study the influence of the entire course of disease on the course of treatment and the long-term treatment related data. A closer look to patients´ charts and the literature was rather disappointing. In most of our patients´ charts the pre-BoNT history was not mentioned or not described in detail. In the literature only little information was available on the course of CD before BoNT therapy. Spontaneous remissions seem to be rare (3 to 61 10%) and usually occur during the first 3 to 5 years after first onset symptoms [14, 22]. In the following years symptoms seem to fluctuate, three different phases of the spontaneous course of CD have been described by neurosurgeons [23]. Deterioration is observed during the first 5 years, stable condition during the next 5 years and possible improvement thereafter [23].

To study the influence of patient´s pre-BoNT history on long-term outcome after BoNT in more detail, a new approach to this problem was used. Patients had to draw the course of disease of CD prior to BoNT therapy (CoDB-graph). Digitization of these graphs allowed to compare and correlate the CoDB-graphs with treatment related data and the long-term outcome of BoNT-therapy. To our knowledge this is the first time that patients´ perception and drawing of the course of disease is used for comparison with physician´s scoring of symptoms and treatment related data.

Please describe all these studies exactly.

Materials and Methods

You said: The present study was performed according to the guidelines for good clinical  practise (GCP) and according to the declaration of Helsinki. It was approved by the local  ethics committee of the University of Düsseldorf (number: 4085).

Please add also the exactl date of the permission for all your experiments.

Please write out in the Materials and Methods section all different groups type. : RO, CO, DO  

Figure 3: please show each CoDB –graph in different colours and note this colours in the Legend.  

Table 2: please add the Abbreviations used like you already in the Table 1.

Conclusions

Please add also the importance of your results for the clinician and how exactly can each clinician use your results every day in the clinical practice.

Author Response

In this study the authors analysed the influence of the course of the disease of idiopathic cervical dystonia (CD) before botulinum toxin therapy (BoNT) on long-term outcomes. For this, 74 CD-patients who were treated on a regular basis in the botulinum toxin outpatient department of the University of Düsseldorf and had received at least 3 injections were consecutively recruited after written in-formed consent. All patients were asked to rate the amount of change of CD in relation to the severity of CD at onset of BoNT therapy (IMPQ). Then they had to draw the course of disease of CD from onset of symptoms until onset of BoNT therapy (CoDB-graph) into a square of 10 × 10 cms. As a result, the course of disease of CD before BoNT therapy has influence on long-term outcome.

The results of the study are new and interesting. Nevertheless, this manuscript needs substantial improvements and corrections before publishing may be possible.

General points:

Please do your References list at the end of the manuscript according to “Toxins”. 

Please add as a Figure an original drawing of 3 different patients.

 Special points:

 Keywords: please add also to keywords: CoDB-graph; patient’s drawing

Introduction

The Introduction section should be substantially improved.

You said:  Cervical dystonia (CD) is the most frequent form of focal dystonia

[1]..

Please add more references at the end of this sentence.

You said: Repetitive intramuscular injections of botulinum toxin (BoNT) every 10 to 16 weeks have become treatment of choice for this chronic neurological disorder [6].

Please add more references at the end of this sentence.

You said:

Nevertheless, up to 46% of BoNT-treated CD-patients discontinue this effective treatment [11]).

Please add more references at the end of this sentence.

You said: It is well-known that CD may progress from a simple type to segmental dystonia [14,15].

Please describe these two studies exactly.

You said: The pattern of complexity of CD may increase with duration of disease even under continuous BoNT treatment [16]. When patients were split-up into a subgroup with an unsatisfactory and into another subgroup of patients with a good effect of BoNT therapy patients with an increase of the pattern of complexity of CD were more frequently found in the subgroup of patients with an “unsatisfactory effect” (for details see [16]). Development from focal to multifocal or segmental dystonia in up to 30% of CD-patients has been reported [14,15], but generalization is a red flag for symptomatic dystonia [17].

Please describe all these studies exactly.

You said: In a recent study on the course of severity of CD it was demonstrated that new symptoms may occur and expand the spectrum of symptoms of CD during BoNT-therapy despite of improvement of head position [18]. Furthermore, it was demonstrated that the duration of the time from onset of symptoms to onset of therapy had a negative impact on long-term BoNT therapy [18]. That was comparable to pallidal deep brain stimulation of primary dystonia where the duration of disease before operation was the only parameter showing a significant (negative) correlation with outcome (for details see [19,20]). 

Furthermore, it has been reported that the initial severity of CD at onset of botulinum toxin therapy has influence on long-term outcome, at least during the first 3 to 4 treatment cycles [21]. But we were not only interested to analyse the influence of the time to therapy or the severity of CD at onset of BoNT therapy on later outcome, but also to study the influence of the entire course of disease on the course of treatment and the long-term treatment related data. A closer look to patients´ charts and the literature was rather disappointing. In most of our patients´ charts the pre-BoNT history was not mentioned or not described in detail. In the literature only little information was available on the course of CD before BoNT therapy. Spontaneous remissions seem to be rare (3 to 61 10%) and usually occur during the first 3 to 5 years after first onset symptoms [14, 22]. In the following years symptoms seem to fluctuate, three different phases of the spontaneous course of CD have been described by neurosurgeons [23]. Deterioration is observed during the first 5 years, stable condition during the next 5 years and possible improvement thereafter [23].

To study the influence of patient´s pre-BoNT history on long-term outcome after BoNT in more detail, a new approach to this problem was used. Patients had to draw the course of disease of CD prior to BoNT therapy (CoDB-graph). Digitization of these graphs allowed to compare and correlate the CoDB-graphs with treatment related data and the long-term outcome of BoNT-therapy. To our knowledge this is the first time that patients´ perception and drawing of the course of disease is used for comparison with physician´s scoring of symptoms and treatment related data.

Please describe all these studies exactly.

Materials and Methods

You said: The present study was performed according to the guidelines for good clinical  practise (GCP) and according to the declaration of Helsinki. It was approved by the local  ethics committee of the University of Düsseldorf (number: 4085).

Please add also the exactl date of the permission for all your experiments.

Please write out in the Materials and Methods section all different groups type. : RO, CO, DO 

Figure 3: please show each CoDB –graph in different colours and note this colours in the Legend. 

Table 2: please add the Abbreviations used like you already in the Table 1.

Conclusions

Please add also the importance of your results for the clinician and how exactly can each clinician use your results every day in the clinical practice.

The references were already placed at the end of the manuscript, but not Materials and Methods. In the revised manuscript we have placed Materials and Methods after the discussion and before the references. 

A new Fig. is added with scans of original drawings.

These keywords are added.

More references are added.

More references are added.

More references are added.

These studies are now described in detail and more references are added.

See above: [16] is cited in detail, [14,15] are described in detail, more references on generalization of dystonias are presented.

The exact date of the approval is given.

The abbreviations RO, CO,DO, OO were explained in Materials and Methods lines and in the section “List of abbreviations”.

We do not intend to publish this manuscript with coloured figures.  

Now we have also added a legend to Table 2.

This point was also raised by reviewer 1 and is addressed in more detail now.

Reviewer 3 Report

Authors did show an interesnting classification and classfication tool. The classification was associated with subjective finding of disease progress. Authors wanted to find the relationship between the disease progress pattern and BTX subscription. To clarify this issue, authors should suppose that all BTX therapy were successful in this study. However, it was not evaluated in this study. 

1. Title was poor and confusing. It should be corrected.
2. In the abstract, dosage and type of BTX should be shown.
3. In the abstract, what's "a square of 10 x 10 cm"? If this was an anatomic site, the exact injection points should be shown.
4. Is "therapy" disease? Why authors did use the term, "onset"? It should be replaced by proper term.
5. Fig. 1, 2, and 3 were "CoDB-graph". Based on this graph, when patient's symptom reached to 100 % severity, patient would receive BTX treatment. In case of Table 1, DURS was 34, 46, and 105 months for RO, CO, and Do group, respectively.  Thus, x-axis of each group should be different. But, they were shown as the same scale in Fig. 2 after normalization. If it was shown as actual duration, Ro and CO group might be shown similar to each other.
6. The reason of similar INDOSE between CO and DO group was unclear considering their significant difference in pre-treatment duration.

Author Response

Authors did show an interesnting classification and classfication tool. The classification was associated with subjective finding of disease progress. Authors wanted to find the relationship between the disease progress pattern and BTX subscription. To clarify this issue, authors should suppose that all BTX therapy were successful in this study. However, it was not evaluated in this study.

1.Title was poor and confusing. It should be corrected.

2. In the abstract, dosage and type of BTX should be shown.

3. In the abstract, what's "a square of 10 x 10 cm"? If this was an anatomic site, the exact injection points should be shown.

4. Is "therapy" disease? Why authors did use the term, "onset"? It should be replaced by proper term.

5. Fig. 1, 2, and 3 were "CoDB-graph". Based on this graph, when patient's symptom reached to 100 % severity, patient would receive BTX treatment. In case of Table 1, DURS was 34, 46, and 105 months for RO, CO, and Do group, respectively.  Thus, x-axis of each group should be different. But, they were shown as the same scale in Fig. 2 after normalization. If it was shown as actual duration, Ro and CO group might be shown similar to each other.

6. The reason of similar INDOSE between CO and DO group was unclear considering their significant difference in pre-treatment duration.

We are thankful for this comment: comparison of ISUI and ATSUI revealed highly significant improvement in all three patient subgroups. This is added now!

 Thanks to reviewer 3 the outcome aspect is emphasized, so that the title is correct now.

A sentence on BoNT/A dose is added.

It is added that the patients had to draw in a sheet of paper.

 We have replaced “onset” by “initiation.

Time scaling does not change curvature. Therefore, graphs of RO- and CO-group patients remain and remained different after normalization according to different treatment durations.

Data are consistent: In the DO-group INDOSE and ADOSE are slightly higher than in the CO-group. Correspondingly ATSUI is slightly better in the DO- than in the CO-group. INDOSE did not correlate with patient´s pre-BoNT history.

We think that treating physicians based their decision which initial dose they chose on at least two factors: duration of pre-treatment and initial severity. This might explain why patient´s in the CO- and DO-group had been injected with nearly the same low initial dose.

This is now emphasized in the discussion.

Round 2

Reviewer 2 Report

Thank you for your corrections. 

This manuscript needs once some before publisching may be possible:

Please format the Refeerences List according to "Toxins".

Lines 34-35: please add more references at the end of this sentzence.

Lines 35-37: please add more references at the end of this sentence.

Lines 238-239: please say: .. (for details see Materials and Methods).

Lines 405-406: please correct this sentence and add the apopriate %.

Author Response

Thank you for your corrections.

This manuscript needs once some before publisching may be possible:

Please format the Refeerences List according to "Toxins".

Lines 34-35: please add more references at the end of this sentzence.

Lines 35-37: please add more references at the end of this sentence.

Lines 238-239: please say: .. (for details see Materials and Methods).

Lines 405-406: please correct this sentence and add the apopriate %.

We are thankful to reviewer 2 for further helpful corrections and comments.

This has been done and cross-checked after addition of further 13 references.

We have added further information and references.

We have added further information and references.

We have performed this correction.

The sentence is corrected and percentage added.